# Initiation of linoleic acid autoxidation with ozone exposure in levitated aerosol particles

Marcel Müller[1,a], Marcel Reichmuth[1], and Ulrich K. Krieger[1]

[1]Institute for Atmospheric and Climate Science, ETH Zurich, Universitätstrasse 16, 8092 Zurich, Switzerland
[a]currently at: Institute of Biogeochemistry and Pollutant Dynamics, ETH Zurich, Universitätstrasse 16, 8092 Zurich, Switzerland

**Correspondence:** Marcel Müller (marcel.mueller@env.ethz.ch)

**Abstract.** When atmospheric aerosol particles undergo oxidation, their physico-chemical properties are altered, influencing their environmental impact. Recent work has revealed that ozonolyis and autoxidation both contribute to the decay of linoleic acid. However, the observed linoleic acid decay rates could only be explained if the autoxidation induction period was shorter in experiments with ozone than without ozone. In this study, we investigate if linoleic acid autoxidation in levitated aerosol particles can be initiated by ozonolysis. Linoleic acid droplets were levitated in an electrodynamic balance, exposed to air with and without ozone, and subsequently analysed by mass spectrometry. Specifically, droplets were first exposed to air containing 39 or 619 ppb ozone for one hour, and thereafter, the gas phase was switched to air without ozone. The observed autoxidation rates were compared with reference experiments without the initial ozonolysis phase, the latter showing an extended induction period. These findings indicate that ozonolysis produces considerable radical concentrations leading to the observed shortening of the induction period. A comparison of the measurements to a simple one-compartment bulk reaction model was used to estimate the radical contribution from ozonolysis. To explain the observed shortening of the induction period, assuming that radicals are formed from the decomposition of linoleic acid oxidation products, a reaction rate constant for the decay of $5.1 \times 10^{-8}$ s$^{-1}$ is obtained. Studies under extreme conditions, such as highly elevated ozone concentrations, may overlook the role of the synergistic effects discussed here, which are only of importance at atmospherically relevant oxidant concentrations.

## 1 Introduction

Aerosol particles undergo ageing in the oxidizing atmosphere, which has an important effect on their role in influencing climate and air quality. Laboratory experiments on model systems, such as levitated particles, have been helpful to determine the impact of aerosol ageing on aerosol particles' physico-chemical properties (e.g., Krieger et al., 2012). Among other proxies, linoleic acid has been used in numerous studies as a model system to study the heterogeneous oxidation of (poly-)unsaturated fatty acids. For example, linoleic acid was found to undergo oxidation through reaction with ozone (Moise and Rudich, 2002; Thornberry and Abbatt, 2004; Broekhuizen, 2004; Hearn and Smith, 2004; Zeng et al., 2013; He et al., 2017; Woden et al., 2024), and because of its bis-allylic hydrogens it is also prone to autoxidation reactions with molecular oxygen (Lee and Chan, 2007; Chu et al., 2019). Therefore, linoleic acid is well-suited for studies on the competition between these two oxidation mechanisms and their synergistic effects, that may ultimately provide valuable insights into the chemical transformations that

aerosol particles undergo in the atmosphere.

In experiments investigating the influence of ozone concentration on linoleic acid oxidation, it was observed that the apparent ozone uptake coefficient depended on the ozone concentration (He et al., 2017; Chu et al., 2019). The higher uptake coefficients at lower ozone concentrations could be explained by a stronger relative contribution of autoxidation (Müller et al.,
2023). Hence, at atmospherically relevant ozone concentrations, the autoxidation with molecular oxygen contributes substantially to linoleic acid degradation. However, in our experiments without ozone, linoleic autoxidation only picked up speed after approximately 40 hours. Yet to explain the observed linoleic acid degradation, autoxidation would have to start earlier – essentially almost at the beginning of the experiment. These recent findings therefore suggest a synergistic effect where the presence of ozone shortens the autoxidation induction period. Radical formation as a result of ozonolysis, as reported in
studies of bulk samples (Goldstein et al., 1968; Pryor et al., 1976), could trigger the autoxidation. For example, Pryor (1994) suggested that approximately 10% of the ozone could react to form radicals. Several mechanisms have been proposed to explain radical formation from oxidation products, including from styrene peroxides (Mayo and Miller, 1956), methyl linoleate hydroperoxides (Morita and Tokita, 2006), linolenate hydroperoxides (Wang et al., 2025), and peroxides from $\alpha$-pinene oxidation (Neuenschwander and Hermans, 2010, 2012). In particular, Kroll et al. (2002) proposed the formation of hydroxyl
radicals from Criegee intermediates in the gas phase, and Zeng et al. (2020) provided evidence for the formation of Criegee intermediates from OH-initiated lipid autoxidation in linoleic acid aerosol particles. In a recent study, Zeng and Wilson (2025) provided further evidence for a linoleic acid aerosol autoxidation mechanism mediated by Criegee intermediates.

In this study, we aim to investigate the influence of different oxidation schemes on the decay of linoleic acid in levitated
droplets. Specifically, we focus on the impact of exposure to ozone prior to oxidation in an atmosphere without ozone, testing both mild (39 ppb) and harsh (619 ppb) ozone concentrations. By examining these conditions, we test our previous hypothesis that ozonolysis can lead to radical formation and initiate autoxidation in levitated linoleic acid droplets. To contextualize our findings, we employ a simple one-compartment bulk reaction model to estimate the contribution of ozone-induced autoxidation initiation.

## 50    2    Instrumentation and methods

The experiments on levitated linoleic acid particles were conducted with electrodynamic balance–mass spectrometry (EDB–MS). The setup used has been introduced previously (Müller et al., 2022). In brief, single linoleic acid droplets were produced from 10 wt-% methanol solutions, charged inductively, and injected into a quadrupolar electrodynamic trap flushed with nitrogen. The droplets were sized, and then exposed to specific gas phases for specific experiment durations. Subsequently, the
particles were ejected into an evaporation unit and the produced gas plume was ionised and analysed with MS.

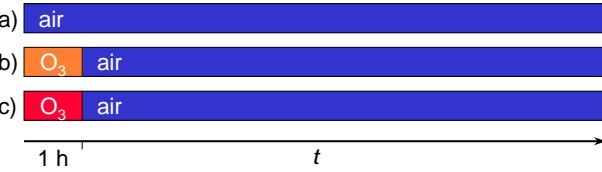

**Figure 1.** Schematic diagram of the different types of experiments conducted: An initial ozonolysis phase of 1 hour duration was applied in experiments (b) and (c), with 39 and 619 ppb ozone, respectively. Ozone was absent in experiments (a) denoted with 'air'.

Linoleic acid ($\geq$ 99%, Sigma-Aldrich) and methanol (UHPLC for MS, Sigma-Aldrich) were used without further purification. Solutions were stored at temperatures below 6 °C and used within one week. Prior to the experiment, levitated droplets were exposed to nitrogen for at least 15 minutes to ensure the complete evaporation of all methanol. The droplets were selected for comparable radii, ranging from 20.1 to 28.8 $\mu$m, with an average $\pm$ standard deviation of 24.1 $\pm$ 2.3 $\mu$m. The size of the droplets was determined by comparing the mean peak-to-peak distance in the 2D angular scattering pattern (captured from 94.87° to 99.13° with a Raspberry Pi HQ Camera V1.0) with simulated spectra (MiePlot v4.6.20, Laven, 2003). The change in size was followed for one levitated droplet per experiment and the observed volume changes were less than 10% in all experiments. The applied gas flux during ageing was 300 sccm, provided from a calibration ozone generator (Photometric $O_3$ Calibrator – Model 401, Advanced Pollution Instrumentation, USA). Given the cell volume of less than 75 cm$^3$, the average residence time of the gas is less than 15 s. The ozone concentration was measured with an electrochemical ozone sensor (Ozone Monitor BMT 932). The nominal concentrations during the initial ozonolysis were 750 and 50 ppb. A better estimate of the concentration in the particle trap was obtained from measurements of the gas outflow concentrations. For the two experiments (a) and (b) (see Figure 1), the measured concentrations within one standard deviation were 619 $\pm$ 25 and 39 $\pm$ 17 ppb. Experiments were conducted at room temperature and under dry conditions (< 15% RH). The temperature of the evaporation unit was set to 190 °C (see Müller et al. (2022)). During MS analysis, a combination of nitrogen flows (1.5 l min$^{-1}$ in total) was used, with 0.3 l min$^{-1}$ applied to transfer the particle to the evaporation unit. Gas-phase analytes were ionised using a cold plasma dielectric barrier discharge ion source (SICRIT SC-30X Ionisation Set, Plasmion GmbH, Germany). Mass spectra were recorded using a triple quadrupole mass spectrometer (QTRAP 4500, AB Sciex LLC, USA) at unit resolution with a scan rate of 2000 Da s$^{-1}$ for data ranges at m/z 84–86, 100–102, 120–122, 142–144, 156–160, 169–173, 184–188, 196–204, 208–212, and 276–283, with a 5 ms pause between mass ranges, resulting in a total scan time of 0.07 s. Mass spectra were integrated over 480 s and corrected for background.

Because of the destructive nature of the analysis, several experiments were conducted with droplets exposed to the same conditions but for different reaction times. Combining data from these experiments allows then to follow the course of the reaction. Experiments were carried out under three different conditions (see Figure 1). In experiments of type (a), droplets were exposed only to air without ozone. The corresponding data set partially includes data already published (Müller et al., 2023). For the other two types of experiments, (b) and (c), linoleic acid droplets were initially exposed to ozone-containing

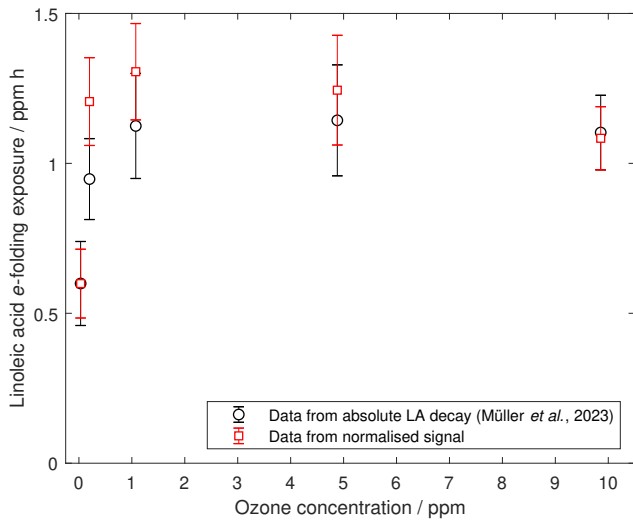

**Figure 2.** $e$-folding exposures of linoleic acid under different ozone concentrations. Original results are compared to results obtained from normalised integrated signals. The calculations of the $e$-folding exposures was carried out as published previously (Müller et al., 2023).

air for one hour, followed by an autoxidation phase in ozone-free air. To ensure the data collected over an extended period
(> 16 months) were comparable, i.e., to account for variability in evaporation, ionisation, and MS-sensitivity, the data were
normalised with respect to the sum of the peaks from linoleic acid (m/z 279) and the most prominent product peak (m/z 171).
Representative spectra from linoleic acid droplets aged with high ozone concentrations, as well as in air without ozone, can
be found in a previously published paper (Fig. 2 in Müller et al., 2023). The linoleic acid fraction was therefore estimated by
dividing the integrated linoleic acid signal by a normalisation divisor $F = \frac{S_{\mathrm{LA}}}{S_{\mathrm{LA},0}} + \frac{S_{171}}{S_{171,\infty}}$ with $S_{\mathrm{LA}}$ denoting the linoleic acid
signal, $S_{171}$ the product signal at m/z 171, $S_{\mathrm{LA},0}$ the average linoleic acid signal from all unreacted droplets, and $S_{171,\infty}$ the
average signal at m/z 171 from all high-exposure experiments with less than 10% linoleic acid remaining. The values used in
this study were $S_{\mathrm{LA},0} = 3.6 \times 10^5$ and $S_{171,\infty} = 2.2 \times 10^5$. Using this normalisation divisor also meant that the measured signal
did not need to be corrected for droplet volume. Normalising the linoleic acid peak was tested on the previously published data
(Müller et al., 2023) and it resulted in comparable linoleic acid decay rates (see Figure 2).

## 95  3   Results and discussion

### 3.1   Qualitative considerations

Data of the linoleic acid decay under the three different experimental types, (a), (b) and (c), are shown in Figure 3. As observed
previously (Müller et al., 2023), the decay of linoleic acid in air reaches the fast radical chain reaction phase only after an
initial induction period of approximately 40 hours. A sigmoid function is fitted to the data from experiments without ozone

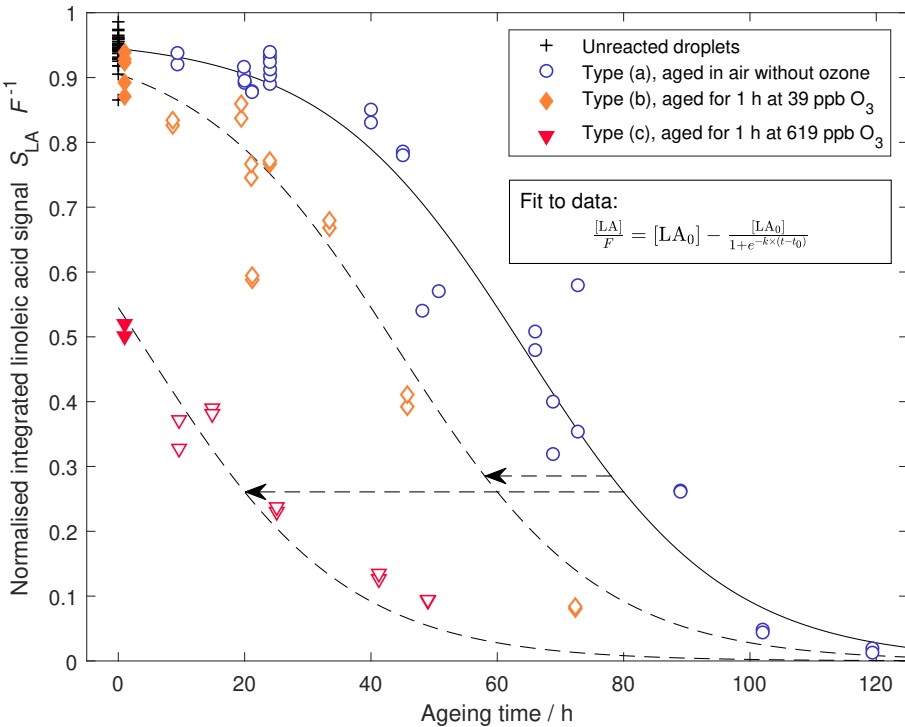

**Figure 3.** Linoleic acid decay with sigmoid fit. Filled symbols represent measurements after initial ozonolysis, and open symbols represent measurements after (subsequent) exposure to air without ozone. Due to the presence of product signal even at zero exposure, the normalised data does not equal one at zero exposure. For the fit (black solid line), only the data from experiments without exposure to ozone exposure was used (+ and ○). The corresponding parameters are $[LA_0] = 0.961$, $k = 0.063$ $h^{-1}$, and $t_0 = 64.3$ h. Two additional sigmoid functions are shown with $t_0 = 44.3$ and $4.3$ h, but otherwise identical parameters (dashed lines).

and displayed as black solid line in Figure 3.

The experiments of type (b) and (c) with initial exposure to ozone show a decrease in linoleic acid concentration during this first hour exposure, the extent of which depends on the ozone concentration. Oxidation with 619 ppb ozone resulted in a decrease of the linoleic acid signal by approximately 50%, whereas oxidation with 39 ppb ozone only led to a reduction of

about 3%. This decay is consistent with the results of our previous study (Müller et al., 2023) and will not be discussed further here. When looking at the autoxidation period past-ozonolysis, it is obvious that linoleic acid continues to decay rapidly after the exposure to 619 ppb ozone, without any noticeable induction period. For the 39 ppb ozone experiments, the ozone phase is still followed by an induction period, which appears to be shortened by approximately 20 hours. Two additional sigmoid functions (dashed lines), with the same parameters as in the fit for experiments of type (a) but shifted by 20 and 60 hours

towards shorter induction periods, are shown to highlight the shortened induction period and the similarities of the oxidation

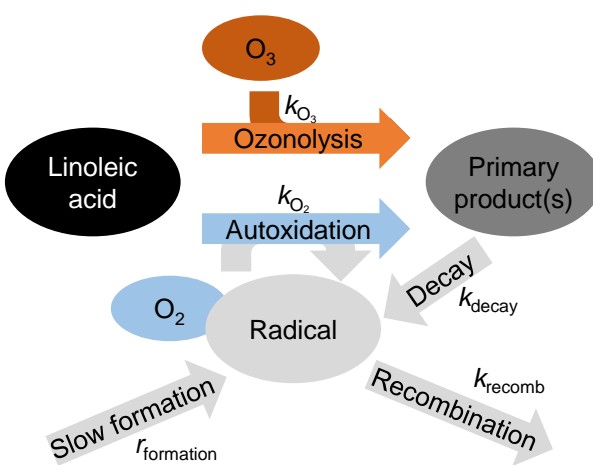

**Figure 4.** Depiction of the reaction scheme with the two linoleic acid decay mechanisms, ozonolysis and autoxidation, and the corresponding reaction rate constants.

decay in air after one hour exposure to 39 or 619 ppb ozone.

Comparing autoxidation rates after ozone-free induction and after ozonolysis allows us to extract the effect of ozonolysis on autoxidation qualitatively. The absence of an induction period in experiments with elevated ozone concentrations (619 ppb) implies a significant concentration of radicals resulting from ozonolysis, which drives the autoxidation after switching to ozone-free air. As the resulting decay rate is comparable to the maximum observed rate in the experiment (a) without ozone exposure, similar radical concentrations are expected in both cases. A similar, but less pronounced effect is observed after exposure to a lower ozone concentration of 39 ppb ozone. The fact that a clear shortening of the induction period is observable even in these experiments implies that the acceleration of autoxidation is highly relevant for atmospheric conditions. Consequently, these results also support our previous assumption (Müller et al., 2023) that an increase in the apparent ozone uptake coefficient towards lower ozone concentrations can be explained with the contribution of autoxidation, particularly because ozonolysis shortens the autoxidation induction period.

## 3.2 Model

To assess the influence of product decay on the radical concentration, we ran a one-compartment process model simulating the bulk chemistry of the linoleic acid system (see Figure 4). We limit ourselves to a simple bulk model because our experiments are limiting the particle sizes (due to the stability regime of the electrodynamic balance and the preferred droplet size for high MS signal), and reaction products cannot be measured. This hinders from an explicit aerosol process model. Using the bulk framework still allowed us to contextualise the collected data. In the model, both autoxidation and ozonolysis are resulting in a decay of the linoleic acid concentration ([LA]). Ozonolysis of unsaturated fatty acids proceeds via the formation of a primary

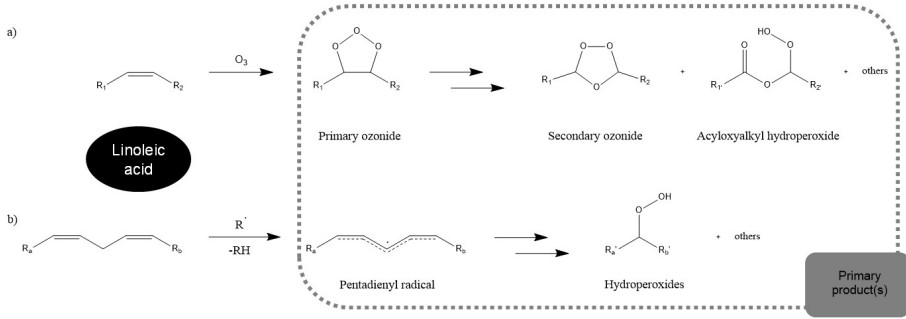

**Figure 5.** Chemical reactions leading to the decay of linoleic acid: a) Ozonolysis and b) autoxidation. Adapted from figure in Müller et al. (2023).

ozonide, which decays rapidly to give an aldehyde and a reactive Criegee intermediate (Criegee, 1975; Moise and Rudich, 2002; Lee and Chan, 2007). The latter are expected to undergo further reactions, such as rearrangements, or reactions with acids and aldehydes, leading to the formation of secondary ozonides and other dimers or oligomers (Figure 5) (e.g., Vesna et al., 2009; Wang et al., 2016; Müller et al., 2022). The autoxidation of linoleic acid occurs mainly by abstraction of the bis-allylic hydrogen, leading to a pentadienyl radical. This can react with oxygen to form a peroxyl radical, which by hydrogen abstraction leads to hydroperoxide formation (Howard and Ingold, 1967; Chu et al., 2019; Müller et al., 2023). For simplicity, the dimers and oligomers from ozonolysis as well as the hydroperoxide products from autoxidation are treated by the model to be identical (termed 'Primary product(s)' in Figure 4). The implications of this assumption are discussed further below in this study.

The initial droplet composition was assumed to be pure linoleic acid, which deviates slightly from the normalised signal of unreacted droplets that accounts for approximately 92% of the total signal. The ozone concentration in linoleic acid, $[O_3]$, was considered as constant and estimated on the basis of a partitioning according to Henry's law for the corresponding solubility in oleic acid of $4.8 \times 10^{-4}$ mol cm$^{-3}$ atm$^{-1}$ (Berkemeier et al., 2021). To the best of our knowledge, the bulk reaction rate of linoleic acid with ozone has not yet been reported in the literature. In our model, the ozonolysis rate ($k_{O_3}$) is used as a fit parameter to match the decay of linoleic acid in experiments with ozone. The simplification of the ozone uptake by using a uniform bulk ozone concentration and Henry's law is expected to be reflected in the resulting ozonolysis rate. However, fitting the reaction scheme to the observations still leads to the measured linoleic acid decay, and to the corresponding product and radical formation. The autoxidation is treated as a function of the radical concentration $[\mathrm{Rad}]$ and to proceed independent of the oxygen concentration, as the hydrogen abstraction reaction (with rate constant $k_{O_2}$) is assumed to be rate-limiting (Maillard et al., 1983), see Equation (1).

$$d/dt[\mathrm{LA}] = -k_{O_2} \times [\mathrm{LA}] \times [\mathrm{Rad}] - k_{O_3} \times [\mathrm{LA}] \times [O_3] \tag{1}$$

For treating the fate of the radicals, an autoxidation rate constant $k_{O_2}$ of $6.2 \times 10^1$ $M^{-1}s^{-1}$ and a radical-radical recombination rate of $4.4 \times 10^6$ $M^{-1}s^{-1}$ were taken from Howard and Ingold (1967). The radical concentration in experiments without ozone must initially increase because of a slow radical formation process, potentially due to impurities in the starting material

that catalyse the oxidation with molecular oxygen. This process is assumed to happen in all experiments, but it is only of relevance during the induction period and in the absence of ozone. In the model, the slow radical formation is implemented with a constant source of radicals represented by a fit parameter ($r_{formation}$) that is expected to be mainly constrained by the experimental data from the induction period. Elucidating the mechanism of radical formation is beyond the scope of this study, so a constant source of radicals is used for simplification. Additionally, radicals are generated during the degradation of

linoleic acid oxidation products, providing a mechanism for the acceleration of autoxidation and for an alternative initiation of autoxidation by ozonolysis. This process is represented by with a unimolecular decay rate ($k_{decay}$), which is used as another fit parameter. Potential reaction schemes for such a degradation have been suggested in the literature (Pryor et al., 1976; Kroll et al., 2002; Morita and Tokita, 2006; Neuenschwander and Hermans, 2012; Zhang et al., 2023; Zeng and Wilson, 2025). These range from simple unimolecular thermal or photochemical decays, such as those of hydroperoxide groups and Criegee

chemistry products, to bimolecular reactions, including reactions with ozone or molecular oxygen. They also include ring-opening reactions of primary ozonides and concerted decompositions of polymeric oxidation products. As capturing specific primary products and pathways is beyond the capability of the analytical methods used in this study, implementing them in our model would only result in unconstrained parameters. For simplicity, the dominant reaction leading to the formation of radicals is assumed to be a unimolecular decay. In this context, our $k_{decay}$ only serves as a proxy for representing all processes that

lead to the formation of new radicals from first-generation products. It disregards the possible bimolecularity of the reaction or reflects it to a certain degree in the form of the reaction rate of the corresponding pseudo-first-order reaction. Complementing the two formation pathways for radicals, autoxidation and ozonolysis, with a radical-radical recombination (with rate $k_{recomb}$), the change in radical concentration ([Rad]) is modelled as stated in Equation (2).

$$d/dt[\text{Rad}] = r_{formation} + k_{decay} \times [\text{Prod}] - 2k_{recomb} \times [\text{Rad}]^2 \tag{2}$$

The relevant concentration of products ([Prod]) is treated to result from the primary oxidation reactions with ozone and molecular oxygen, see Equation (3).

$$d/dt[\text{Prod}] = k_{O_2} \times [\text{LA}] \times [\text{Rad}] + k_{O_3} \times [\text{LA}] \times [\text{O}_3] - k_{decay} \times [\text{Prod}] \tag{3}$$

This set of equations, while a simplification, serves to estimate the impact of ozonolysis on autoxidation. The use of only one product species and the representation of all possible product decay reactions with one reaction rate are supported by the

observation that autoxidation rates post-ozonolysis under high ozone concentrations are similar to the autoxidation rates in experiments with no ozone (see discussion below). The used parameter values and the allowed ranges are provided in Table 1.

To infer the parameter values that best describe the measured linoleic acid decay, we used Markov chain Monte Carlo (MCMC) simulations. The compositional evolution over time was simulated with an ordinary differential equation solver in

**Table 1.** Model parameters for the bulk model (fit parameters in the upper part and literature values in the lower part of the table).

| Parameter | Allowed range | Reference |
|---|---|---|
| Ozonolysis rate constant $k_{O_3}$ [$M^{-1}\,s^{-1}$] | $[10^2 - 10^4]$ | Müller et al. (2023)[†] |
| Slow radical formation rate $r_{formation}$ [$M\,s^{-1}$] | $[10^{-20} - 10^{-1}]$[‡] | |
| Product decay rate constant $k_{decay}$ [$s^{-1}$] | $[10^{-10} - 10^{-6}]$ | |
| Autoxidation rate constant $k_{O_2}$ [$M^{-1}\,s^{-1}$] | $6.2 \times 10^1$ | Howard and Ingold (1967) |
| Radical recombination rate constant $k_{recomb}$ [$M^{-1}\,s^{-1}$] | $4.4 \times 10^6$ | Howard and Ingold (1967) |

[†] Estimated from their linoleic acid decay at 10 ppm ozone and applying Henry's law as described in the current study.

[‡] On the lower side, the radical formation rate is limited by the rate required to generate at least one radical in a droplet of radius 25 $\mu$m within the first 40 h of the experiment, which corresponds to an $r_{formation}$ of approximately $3 \times 10^{-20}\,M\,s^{-1}$.

**Table 2.** Model parameter values for the global fit (fit parameters in the upper part and literature values in the lower part of the table). The 95% intervals, provided in brackets, are based on the MCMC sampling.

| Parameter | Global fit |
|---|---|
| Ozonolysis rate constant $k_{O_3}$ [$M^{-1}\,s^{-1}$] | $7.3 \times 10^2$ ($5.9 \times 10^2$, $8.9 \times 10^2$) |
| Slow radical formation rate $r_{formation}$ [$M\,s^{-1}$] | $\lesssim 6.3 \times 10^{-10}$ |
| Product decay rate constant $k_{decay}$ [$s^{-1}$] | $5.1 \times 10^{-8}$ ($4.4 \times 10^{-8}$, $5.9 \times 10^{-8}$) |
| Autoxidation rate constant $k_{O_2}$ [$M^{-1}\,s^{-1}$] | $6.2 \times 10^1$ (Howard and Ingold, 1967) |
| Radical recombination rate constant $k_{recomb}$ [$M^{-1}\,s^{-1}$] | $4.4 \times 10^6$ (Howard and Ingold, 1967) |

MATLAB R2019a with relative error tolerances of $1 \times 10^{-6}$ and absolute error tolerances of $1 \times 10^{-12}$. The MCMC simulations were carried out using UQLab version 2.0 (Marelli and Sudret, 2014; Wagner et al., 2019). The fit parameters were sampled with log-uniform prior distribution using the ranges specified in Table 1. The allowed range for the ozonolysis and product decay rates were manually adjusted to capture more than the range required to reproduce the decay curve characteristics. For the experimentally determined values, a Gaussian discrepancy term was assumed with a relatively big standard error of 0.1 times the highest signal to be on the safe side in the event of any errors introduced from the standardisation calculation. The MCMC was run with $1 \times 10^5$ steps, of which the first $5 \times 10^4$ were discarded to avoid burn-in artefacts. The corresponding findings are summarised in Table 2.

While for the ozonolysis and for the product decay rate, single values can be extracted, the slow radical formation rate cannot be well constrained. This is because for values lower than approximately $6.3 \times 10^{-10}$ M s$^{-1}$, the model is not sensitive to changes in the slow radical formation rate, as the total radical formation is dominated by the faster radical formation from autoxidation products. The slow radical formation would only have a significant impact on the outcome when it is assumed to be high enough to shorten the induction period in experiments without ozone, which would not be compatible with the obser-

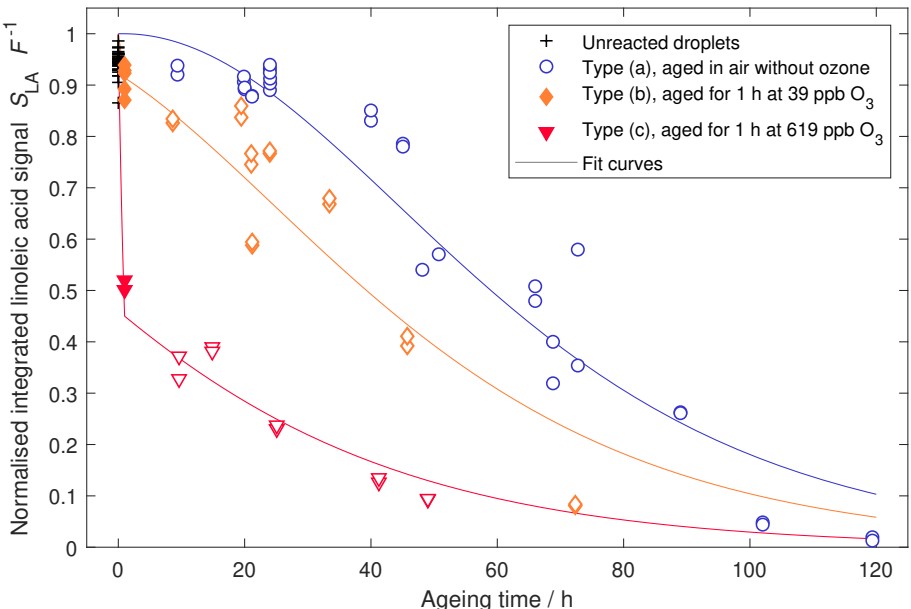

**Figure 6.** Experimental results and model curves resulting from the MCMC point estimate (posterior mean) for the model parameters as given in Table 2. Filled symbols represent measurements after initial ozonolysis, and open symbols represent measurements after (subsequent) exposure to air without ozone.

vations. When the obtained posterior mean values for the parameters are plugged into the model ($k_{O_3} = 7.3 \times 10^2$ M$^{-1}$s$^{-1}$,

$r_{formation} = 1.1 \times 10^{-16}$ M s$^{-1}$, and $k_{decay} = 5.1 \times 10^{-8}$ s$^{-1}$), the model captures the data rather well (Figure 6). Notably, the model reflects the observed shortening of the induction period and the subsequent rapid decay phase for conditions with ozone, while resulting in a sigmoid shape for ageing in the absence of ozone. However, the model cannot be brought into perfect agreement with all experimental data. The obtained point estimates lead to a slight overestimation of the ozonolysis progress, and the linoleic acid concentrations in the low ozone condition experiments are underestimated for most of the experiments.

These deviations may be due to the simplified model, which treats the potentially complex mixture of products and their ability to form radicals with one single parameter. An estimation for the separated contributions to autoxidation can be obtained when the MCMC simulation is run on a subset of the experimental data. For example, when using only the high-ozone data or only the data from experiments with no exposure to ozone. The corresponding simulations result in comparable product decay rates, which suggests that the different oxidation products do not differ substantially in their ability to cause autoxidation (see Table 3

and Figure A1). In any case, it can be concluded that for our simple one-compartment bulk model, product decay rates on the order of $10^{-8}$ s$^{-1}$ are required to reproduce the experimentally observed linoleic acid decay, namely a substantial shortening of the autoxidation induction period in the presence of low concentrations of ozone.

**Table 3.** Inferred model parameter values (posterior mean) for the additional fits in comparison to the global fit. The 95% intervals, provided in brackets, are based on the MCMC sampling.

| Parameter | Global fit | 619 ppb ageing | Ozone-free air ageing |
|---|---|---|---|
| Ozonolysis rate constant $k_{O_3}$ [M$^{-1}$ s$^{-1}$] | $7.3 \times 10^2$ $(5.9 \times 10^2, 8.9 \times 10^2)$ | $6.4 \times 10^2$ $(4.5 \times 10^2, 8.9 \times 10^2)$ | – |
| Slow radical formation rate $r_{formation}$ [M s$^{-1}$] | $\lesssim 6.3 \times 10^{-10}$ | $\lesssim 3.5 \times 10^{-7}$ | $\lesssim 1.4 \times 10^{-9}$ |
| Product decay rate constant $k_{decay}$ [s$^{-1}$] | $5.1 \times 10^{-8}$ $(4.4 \times 10^{-8}, 5.9 \times 10^{-8})$ | $6.8 \times 10^{-8}$ $(1.3 \times 10^{-8}, 2.0 \times 10^{-7})$ | $4.8 \times 10^{-8}$ $(4.1 \times 10^{-8}, 5.6 \times 10^{-8})$ |

It should be noted that early oxidation products, such as slightly oxidised monomers, could strongly differ in terms of their decay and radical formation potential from later oxidation products that are more oxidised and potentially larger oligomers. Because of the thermal instability of dimers and oligomers, our instrumental setup does not allow us to distinguish the different types of products (Müller et al., 2023). Therefore, we were not able to capture differences in oxidation products, and their potentially different contributions to autoxidation are averaged in our approach presented here. The reaction system understanding could potentially be significantly improved if product formation was integrated in the analysis.

When applying the data presented here to atmospheric particles, it is important to take into account differences in both particle composition and size. In the atmosphere, particles are much smaller than in our experiment (Rose et al., 2021). If autoxidation is assumed to be a bulk process, variations in particle size would not directly affect the autoxidation process and therefore would not affect our results after ozonolysis. However, ozonolysis is significantly affected by particle size, as ozone uptake depends on the ratio of surface area to volume (Zeng et al., 2013). Consequently, it would be expected that the induction phase of autoxidation becomes even shorter for smaller particles, and the relative contribution of ozonolysis increases overall. Further experiments on other particle sizes and with the possibility to quantify products would be desirable to advance the understanding of this reaction system and its implications.

## 4 Conclusions

In this study, we investigated the effect of ozone on autoxidation in levitated linoleic acid particles. Our data support the hypothesis that under low ozone conditions, ozonolysis can contribute to the initiation of autoxidation in linoleic acid droplets. This effect may likely be overlooked when experiments are carried out at higher ozone concentrations and over shorter reaction times than are atmospherically relevant. The types of experiments presented in this study may help to distinguish between the effects of ozonolysis and autoxidation. While the ozonolysis and autoxidation chemistry of linoleic acid is complex, our simple model is able to reproduce the main observed trends in linoleic acid decay while representing additional radical formation with

one decay rate for oxidation products (estimated at $5.1 \times 10^{-8}$ s$^{-1}$). More experiments with different detection techniques and other sequences of gas-phase exposures may help to disentangle the complex processes happening in linoleic acid droplets, and in atmospheric aerosol oxidation in general. Synergistic effects can play important roles at atmospherically relevant oxidant concentrations and may need to be considered, when investigating atmospheric processes and their impacts. Therefore, choosing adequate reaction conditions that capture these processes is required to interpret the impacts of synergistic effects.

*Data availability.* The ageing times and the normalised integrated linoleic acid signal of the data points in Figures 3 and 6 are available online at https://doi.org/10.3929/ethz-b-000727019 (Müller et al., 2025).

## Appendix A: Model

Figure A1 presents the modelled concentrations of the three species for each experiment type, based on simulations using the posterior mean values from the MCMC analyses. In the case where only the 619 ppb ozone data are used, the model places greater emphasis on the ozone-driven decay. This results in a lower estimated $k_{O_3}$ and a higher $k_{\text{decay}}$, compared to the other model runs, in order to best fit the data. Despite these differences, the overall concentration profiles are relatively similar across conditions, and the modelled linoleic acid concentrations are in good agreement with the observations.

*Author contributions.* Funding was acquisited by U. K. K. Experiments were conceptualised by M. M., and carried out by M. M. and M. R., the data was analysed by M. M. and M. R. The results were discussed by M. M., M. R., and U. K. K. The manuscript was written by M. M. and revised by M. R. and U. K. K.

*Competing interests.* No competing interests are present.

*Acknowledgements.* This work was supported by ETH Research Grant ETH-03 17-2. Microsoft 365 Copilot was used to improve phrasing and paragraph structures. DeepL Write was used to improve phrasing. Both tools were accessed between December 2024 and June 2025. The authors would like to thank the two anonymous reviewers for their feedback.

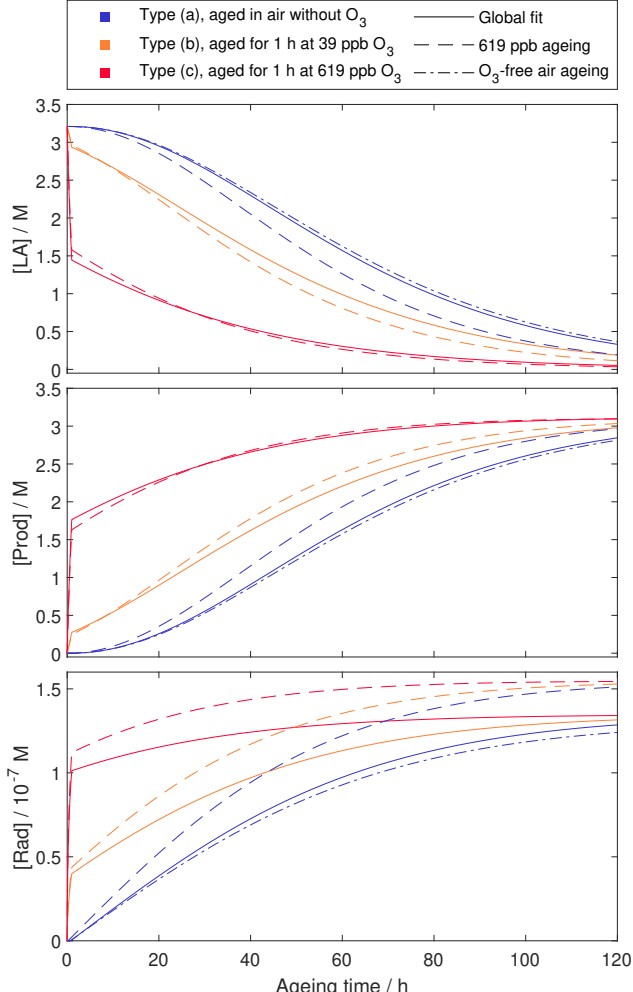

**Figure A1.** Model concentrations of linoleic acid ([LA]), products ([Prod]), and radicals ([Rad]) for each of the three experiment types that were modelled. Note that for the model run without ozone, no ozonolysis rate was determined and therefore only the model run without ozone is shown for this model.

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
