# Peer review of "Initiation of linoleic acid autoxidation with ozone exposure in levitated aerosol particles"

_EGUsphere, 2025_

## Author Response (AR1)

We thank both anonymous reviewers for their critical reviews and constructive comments. We believe they helped to improve the paper.

Below, we address the reviewers comments (reprinted in bold) with answers (in normal font) including excerpts from the new version (in italics).

**RC1:**

**This manuscript presents kinetics of multiphase linoleic acid aging in the presence of oxygen and ozone to investigate the importance of ozone for initiating autoxidation processes. The experiments use a novel method that can follow the decay of linoleic acid in micron-sized droplets for up to 5 days. Within the limitations of the measurement, which does not provide insight into changes in reaction products between oxygen and ozone experiments, the authors provide evidence for ozone shortening the induction period for autoxidation and therefore that ozone chemistry likely generates radicals in the condensed phase. This work has implications for our understanding of lipid oxidation across a range of environments and applications.**

**Overall, the manuscript is clear in its descriptions and well-written, if a bit thin on explanations and discussion. The description of the experiments and approach is sufficiently complete, given that the authors rely significantly on a previous paper (Müller et al., 2023)**

We expanded the explanations and discussions in some instances, where deemed fit. All changes are discussed below.

**This manuscript it suitable for publication after the following minor comments are addressed.**

**Major Comments:**

**1. Given the brevity of this paper, the information in the appendix could be easily incorporated into the main text. This will help the reader get the details as they read, rather than having to flip to the appendix.**

Agreed, we incorporated the appendix into the main text and adjusted the text to maintain the reading flow.

**Specific Comments:**

**1. L39-41: It is worth discussing further that OH formation from carbonyl oxides has only been shown clearly in the gas phase (Kroll 2002), while the second paper discussed in this sentence (Zeng 2020) is a multiphase OH-oxidation experiment showing evidence for carbonyl oxides generated by OH chemistry in aerosol. Zeng 2020 proposes that carbonyl oxides may decompose to form an alkoxy radical and OH in the condensed phase.**

This is a good clarification. We edited the text by adding '*in the gas phase*' for the first part. In addition, we also added the reference to a very recent publication with the following sentence:

'*In a recent study, Zeng and Wilson (2025) provided further evidence for a linoleic acid aerosol autoxidation mechanism mediated by Criegee intermediates.*'

**2. L115: And secondary ozonides under dry conditions?**

Yes, this is correct. As secondary ozonides are a product of two smaller early products, we included them in the originally used expression 'dimers'. We adapted the original sentence to better represent the complexity of the chemistry. It now reads:

*'The latter are expected to undergo further reactions, such as rearrangements, or reactions with acids and aldehydes, leading to the formation of secondary ozonides and other dimers or oligomers (Figure 5) (e.g., Vesna et al., 2009; Wang et al., 2016; Müller et al., 2022).'*

**3. L121: Presumably the Henry's Law constant (i.e. air-water partition constant) is not used to model the bulk ozone concentration, but rather an air-linoleic acid partition constant for ozone? This is later discussed in the appendix, but would more suitably be discussed here.**

We moved the whole discussion of the representation of the ozone concentration from the appendix to this part of the discussion. It now reads as follows:

*'The ozone concentration in linoleic acid, $[O_3]$, was considered as constant and estimated on the basis of a partitioning according to Henry's law for the corresponding solubility in oleic acid of $4.8 \times 10^{-4}$ mol $cm^{-3}$ $atm^{-1}$ (Berkemeier et al., 2021). To the best of our knowledge, the bulk reaction rate of linoleic acid with ozone has not yet been reported in the literature. In our model, the ozonolysis rate ($k_{O3}$) is used as a fit parameter to match the decay of linoleic acid in experiments with ozone. The simplification of the ozone uptake by using a uniform bulk ozone concentration and Henry's law is expected to be reflected in the resulting ozonolysis rate. However, fitting the reaction scheme to the observations still leads to the measured linoleic acid decay, and to the corresponding product and radical formation.'*

**4. L129 & 132: What fit parameter? Fit to what? This discussion of the model appears insufficient in the main text, and the reader must go to the appendix to get pertinent information.**

In this case, too, the information from the appendix was transferred to the main body of the paper, and the passage in question was edited as follows:

*'In the model, the slow radical formation is implemented with a constant source of radicals represented by a fit parameter ($r_{formation}$) that is expected to be mainly constrained by the experimental data from the induction period. Elucidating the mechanism of radical formation is beyond the scope of this study, so a constant source of radicals is used for simplification. Additionally, radicals are generated during the degradation of linoleic acid oxidation products, providing a mechanism for the acceleration of autoxidation and for an alternative initiation of autoxidation by ozonolysis. This process is represented by with a unimolecular decay rate ($k_{decay}$), which is used as another fit parameter.'*

**5. L132-133: This is relying very heavily on literature -- the authors should provide a summary relevant to their experiment here.**

Thank you also for this suggestion. We added a brief description of the chemistry suggested in the literature and add how we arrive at representing the potentially complex mixture of products and therefore numerous conceivable reactions with one single reaction rate. This reads now:

*'Potential reaction schemes for such a degradation have been suggested in the literature (Pryor et al., 1976; Kroll et al., 2002; Morita and Tokita, 2006; Neuenschwander and Hermans, 2012; Zhang et al., 2023; Zeng and Wilson, 2025). These range from simple unimolecular thermal or photochemical decays, such as those of hydroperoxide groups and Criegee chemistry products, to bimolecular reactions, including reactions with ozone or molecular oxygen. They also include ring-opening reactions of primary ozonides and concerted decompositions of polymeric oxidation products. As capturing specific primary products and pathways is beyond the capability of the analytical methods used in this study, implementing them in our model would only result in unconstrained parameters. For simplicity, the dominant reaction leading to the formation of radicals is assumed to be a unimolecular decay. In this context, our $k_{decay}$ only serves as a proxy for representing all processes that*

*lead to the formation of new radicals from first-generation products. It disregards the possible bimolecularity of the reaction or reflects it to a certain degree in the form of the reaction rate of the corresponding pseudo-first-order reaction.*'

**6. L146 & Table 1: Has the bulk k_O3 for linoleic acid measured in the literature? If not, the lack of constraint is work discussing.**

To the best of our knowledge, bulk reaction rates for the ozonolysis of linoleic acid have not yet been measured. We added this information at the beginning of section 3.2, when the treatment of the ozonolysis in the model is introduced:

'*To the best of our knowledge, the bulk reaction rate of linoleic acid with ozone has not yet been reported in the literature. In our model, the ozonolysis rate ($k_{O3}$) is used as a fit parameter to match the decay of linoleic acid in experiments with ozone.*'

**RC2:**

**In the manuscript "Initiation of linoleic acid autoxidation with ozone exposure in levitated aerosol particles", Müller et al. present data showing that autooxidation of linoleic acid is accelerated in microdroplets that have been previously exposed to ozone than in ozone free droplets. They construct a simple one-component model to explain these results whereby oxidation products of ozonolysis/autooxidation slowly react to form radicals that can promote autooxidation. Overall, the synergistic effect between ozonolysis and autooxidation is interesting, but the manuscript is rather 'data-light' and there are issues with the chemical model that make it difficult to see what the broad implications of synergistic effects between ozonolysis and autooxidation are in the atmosphere. I've highlighted several areas the authors should clarify their assumptions and suggested other data to be included below. After addressing these comments and suggestions, I believe the manuscript is suitable for publication in Atmospheric Chemistry and Physics.**

Thank you for your comments. As explained below, we have now tried to better emphasise the experimental nature of the work so that the broader implications can be better understood.

**Instrumentation: The authors should provide more details on their experimental setup, including: How are droplets sized? What is the typical residence of gases in the EDB given their experimental flow rates? Are droplet sizes measured throughout the measurement and do they change with droplet age?**

The experimental setup has been described in detail in previous publications (Müller et al. (2022) and Müller et al. (2023)). However, we see the need for more details in order to fully comprehend the experiments and their results. We expanded the description of the experimental technique in order to cover the questions raised:

'*The size of the droplets was determined by comparing the mean peak-to-peak distance in the 2D angular scattering pattern (captured from 94.87° to 99.13° with a Raspberry Pi HQ Camera V1.0) with simulated spectra (MiePlot v4.6.20, Laven, 2003). The change in size was followed for one levitated droplet per experiment and the observed volume changes were less than 10% in all experiments. The applied gas flux during ageing was 300 sccm, provided from a calibration ozone generator (Photometric O3 Calibrator – Model 401, Advanced Pollution Instrumentation, USA). Given the cell volume of less than 75 cm³, the average residence time of the gas is less than 15 s.*'

**Figures 2 and 4 appear to show identical data. If the sigmoidal fits are purely empirical and not derived from theory, they should be omitted. The model results shown in Figure 4 can used to qualitatively describe the observed trends.**

We agree that the data points in both figures are identical. However, we would like to refrain from removing the graph with the sigmoid fits, because we are convinced that one of the key findings of our study, namely the shortened induction period, can be shown very well using the sigmoid fits alone without relying on any modelling. Figure 2 thus supports the qualitative discussion of our results very well without relying on the assumptions made in the model.

**It would be useful to include structures for the chemistry that is being described to aid those non-chemistry-inclined readers (e.g., abstraction of bis-allylic hydrogen)**

We added a new figure (Figure 5 in the new version) with the first steps for both reaction types to help the reader. However, we do not go into further detail with this mechanism, as our instrumental setup does not allow us to distinguish between the different products.

**The model is overly simplistic and some of the assumptions are not well explained/constrained to make it useful beyond this system here. For example, the role of oxygen in producing radicals is not included. Most likely, the small droplet sizes (~25 micron radius) facilitate radical formation by preventing oxygen limitations. However, this isn't thoroughly addressed by the authors. Do they expect droplets of different sizes to show different kinetics? It would be useful to perform similar experiments in droplets of different sizes (and/or in a macroscale quantity of linoleic acid) to assess the role of oxygen limitations or other surface mediated effects.**

The role of droplet size and the comparison to bulk experiments are indeed an interesting suggestion. With our current setup, we are, however, limited to a narrow range of acceptable droplet sizes. More than a factor of two in droplet size is hardly possible due to the stability regime of our electrodynamic balance and the used droplet sizes are preferred for highest MS signal-to-noise ratios. Additionally, our sizing method comes with an uncertainty of approximately 10% for radius detection, limiting the significance of size-dependent experiments. Consequently, with our experiment and our data we do not claim to close the gap between experimental and explicit chemical models. Our aim here is to contextualise the collected data and make it plausible.

We edited the text at the beginning and at the end of section 3.2 to make clearer what the (experimental) limitations are and what our goals are:

*'To assess the influence of product decay on the radical concentration, we ran a one-compartment process model simulating the bulk chemistry of the linoleic acid system (see Figure 4). We limit ourselves to a simple bulk model because our experiments are limiting the particle sizes (due to the stability regime of the electrodynamic balance and the preferred droplet size for high MS signal), and reaction products cannot be measured. This hinders from an explicit aerosol process model. Using the bulk framework still allowed us to contextualise the collected data.'*

and

*'Further experiments on other particle sizes and with the possibility to quantify products would be desirable to advance the understanding of this reaction system and its implications.'*

**Line 126: "The radical concentration in experiments without ozone must initially increase because of a slow radical formation process, potentially due to impurities in the starting material that catalyse the oxidation with molecular oxygen…. In the model, the slow radical formation is implemented with a constant source of radicals represented by a fit parameter." It is unclear why a constant rate of radical formation is used here and not a rate constant for reaction of linoleic acid. As to the point above, the authors should comment on how this rate (or rate constant) is expected to change with different oxygen concentrations and droplet sizes.**

We agree that from the point of view of fundamentally understanding the chemical system, it would be interesting to learn what the underlying chemical reactions are and how the initial formation of radical depends on other factors. However, due to the limited ability to resolve the detailed mechanism with our experimental setup, we are not able to infer more details about this process. Because of the autocatalytic nature of autoxidation, the initial slow radical formation is only of relevance during the induction period and is later outranged by the autoxidative radical formation. Therefore, it is impossible for us to rule out a specific reaction mechanism. Assuming a stochastic process that eventually leads to the first radicals was chosen for the sheer simplicity of the assumption. The limited explanatory power of the data from the induction period is also reflected in the fact that no single $r_{formation}$ could be determined, but only a range of plausible values was obtained.

We edited the text to make this clearer:

'*The radical concentration in experiments without ozone must initially increase because of a slow radical formation process, potentially due to impurities in the starting material that catalyse the oxidation with molecular oxygen. This process is assumed to happen in all experiments, but it is only of relevance during the induction period and in the absence of ozone. In the model, the slow radical formation is implemented with a constant source of radicals represented by a fit parameter ($r_{formation}$) that is expected to be mainly constrained by the experimental data from the induction period. Elucidating the mechanism of radical formation is beyond the scope of this study, so a constant source of radicals is used for simplification.*'

**Line 130: "Additionally, radicals are generated during the degradation of linoleic acid oxidation products, providing a mechanism for the acceleration of autoxidation and for an alternative initiation of autoxidation by ozonolysis. This unimolecular process is represented by another fit parameter." The authors need to further justify their choice of a unimolecular rate constant for the production of radicals from primary products. Many of the reaction schemes suggested in the citations (e.g., Neuenschwander and Hermans, 2012) propose bimolecular rate constants for the production of radical species from non-radical reactants.**

Again, from our set of data it is impossible to draw final conclusions regarding the detailed chemical mechanism of autoxidation. We interpret the autoxidative decay of linoleic acid as a complex interplay of several reactions. Yet, it is interesting to note that already assuming one single unimolecular reaction rate suffices to capture the decay of linoleic acid relatively well. We expanded the discussion in the paper to give more context, but we stick to the first-order rate, which could be interpreted as pseudo-first order rate, to keep the system as simple as possible:

'*Additionally, radicals are generated during the degradation of linoleic acid oxidation products, providing a mechanism for the acceleration of autoxidation and for an alternative initiation of autoxidation by ozonolysis. This process is represented by with a unimolecular decay rate ($k_{decay}$), which is used as another fit parameter. Potential reaction schemes for such a degradation have been suggested in the literature (Pryor et al., 1976; Kroll et al., 2002; Morita and Tokita, 2006; Neuenschwander and Hermans, 2012; Zhang et al., 2023; Zeng and Wilson, 2025). These range from simple unimolecular thermal or photochemical decays, such as those of hydroperoxide groups and Criegee chemistry products, to bimolecular reactions, including reactions with ozone or molecular oxygen. They also include ring-opening reactions of primary ozonides and concerted decompositions of polymeric oxidation products. As capturing specific primary products and pathways is beyond the capability of the analytical methods used in this study, implementing them in our model would only result in unconstrained parameters. For simplicity, the dominant reaction leading to the formation of radicals is assumed to be a unimolecular decay. In this context, our $k_{decay}$ only serves as a proxy for*

*representing all processes that lead to the formation of new radicals from first-generation products. It disregards the possible bimolecularity of the reaction or reflects it to a certain degree in the form of the reaction rate of the corresponding pseudo-first-order reaction.'*

**Line 155 "The corresponding simulations result in comparable product decay rates, which suggests that the different oxidation products do not differ substantially in their ability to cause autoxidation (see Table B2 in the appendix)." While the product decay rate constants are similar, the 'slow' radical formation rate for these two cases (619 ppb and ozone free) are significantly different. They are also both significantly larger than the 'global fit' radical formation rate. Presumably, if the model were accurately reflecting the chemistry in the microdroplets, fit rate constants for the subset of experimental data should be closer to the global fit (within error) and not $10^3$ times larger. The authors need to address these differences in best fit 'slow' radical formation rate for the two subsets.**

The limits for $r_{formation}$ in Table 3 (new version) are determined mainly by the linoleic acid decay slope at low turnover, i.e. at autoxidation induction. Because only the case without ozone shows an extended induction period, the slow radical formation is most restricted here. Obtaining higher possible values for the experiments with ozone exposure is not a contradiction but a logical consequence of less restriction due to the underlying data. Please note that the values in Table 2 and 3 (new version) are not given as posterior mean with confidence intervals, but rather as upper boundaries from the posterior distribution. Only for the fit in Figure 6, we chose the mean posterior value for the representative model curve.

We refrain from repeating the information in the respective paragraph in order to avoid moving the attention from the autocatalytic radical formation. We hope the matter is clearer with our new formulation in the preceding paragraphs:

*'The radical concentration in experiments without ozone must initially increase because of a slow radical formation process, potentially due to impurities in the starting material that catalyse the oxidation with molecular oxygen. This process is assumed to happen in all experiments, but it is only of relevance during the induction period and in the absence of ozone. In the model, the slow radical formation is implemented with a constant source of radicals represented by a fit parameter ($r_{formation}$) that is expected to be mainly constrained by the experimental data from the induction period.'*

**The authors should provide a plot of the concentrations of each of the species in the model over time for each of the different types of reactions. This information can be provided in the SI.**

We added a Figure (A1) with the three concentrations of linoleic acid, the primary products and radicals for each of the three experiment types that were modelled.

*'Figure A1 presents the modelled concentrations of the three species for each experiment type, based on simulations using the posterior mean values from the MCMC analysis. In the case where only the 619 ppb ozone data are used, the model places greater emphasis on the ozone-driven decay. This results in a lower estimated $k_{O3}$ and a higher $k_{decay}$, compared to the other model runs, in order to best fit the data. Despite these differences, the overall concentration profiles are relatively similar across conditions, and the modelled linoleic acid concentrations are in good agreement with the observations.'*

**Conclusions: The authors should describe in more detail how their results showing synergistic effects are relevant to other more atmospherically relevant systems and what the implications of the effects are.**

We edited the conclusions to highlight better our point about many laboratory experiments missing potentially relevant effects due to deviations from atmospheric conditions and the nature of our study as an exploratory study to separate the effects of different reaction mechanisms:

'*Our data support the hypothesis that under low ozone conditions, ozonolysis can contribute to the initiation of autoxidation in linoleic acid droplets. This effect may likely be overlooked when experiments are carried out at higher ozone concentrations and over shorter reaction times than are atmospherically relevant. The types of experiments presented in this study may help to distinguish between the effects of ozonolysis and autoxidation.*'

and

'*Synergistic effects can play important roles at atmospherically relevant oxidant concentrations and may need to be considered, when investigating atmospheric processes and their impacts. Therefore, choosing adequate reaction conditions that capture these processes is required to interpret the impacts of synergistic effects.*'

**Appendix A. The authors should include representative mass spectra to justify their choice of normalization scheme.**

We thank you for the comment and insert the sentence below. As representative data has already been published, reference is only made here to our previous study. The figure (new Figure 2) showing the effect of normalisation has been moved from the appendix to section 2 of the current study.

'*Representative spectra from linoleic acid droplets aged with high ozone concentrations, as well as in air without ozone, can be found in a previously published paper (Fig. 2 in Müller et al., 2023).*'